# Smoking-by-genotype interaction in type 2 diabetes risk and fasting glucose

Peitao Wu[1]*, Denis Rybin[1], Lawrence F. Bielak[2], Mary F. Feitosa[3], Nora Franceschini[4], Yize Li[5], Yingchang Lu[6], Jonathan Marten[7], Solomon K. Musani[8], Raymond Noordam[9], Sridharan Raghavan[10,11,12], Lynda M. Rose[13], Karen Schwander[5], Albert V. Smith[14,15], Salman M. Tajuddin[16], Dina Vojinovic[17], Najaf Amin[17], Donna K. Arnett[18], Erwin P. Bottinger[6], Ayse Demirkan[17], Jose C. Florez[19,20,21], Mohsen Ghanbari[17,22], Tamara B. Harris[23], Lenore J. Launer[23], Jingmin Liu[24], Jun Liu[17], Dennis O. Mook-Kanamori[25,26], Alison D. Murray[27], Mike A. Nalls[28,29], Patricia A. Peyser[2], André G. Uitterlinden[30], Trudy Voortman[17], Claude Bouchard[31], Daniel Chasman[13,32], Adolfo Correa[33], Renée de Mutsert[25], Michele K. Evans[16], Vilmundur Gudnason[34,35], Caroline Hayward[7], Linda Kao[36,37,38]†, Sharon L. R. Kardia[2], Charles Kooperberg[24], Ruth J. F. Loos[6,39], Michael M. Province[3], Tuomo Rankinen[31], Susan Redline[32,40,41], Paul M. Ridker[13,32], Jerome I. Rotter[42], David Siscovick[43], Blair H. Smith[44], Cornelia van Duijn[17], Alan B. Zonderman[16], D. C. Rao[45], James G. Wilson[33], Josée Dupuis[1,46], James B. Meigs[20,47,48], Ching-Ti Liu[1]*, Jason L. Vassy[48,49]*

1 Department of Biostatistics, Boston University School of Public Health, Boston, Massachusetts, MA, United States of America, 2 Department of Epidemiology, School of Public Health, University of Michigan, Ann Arbor, MI, United States of America, 3 Division of Statistical Genomics, Department of Genetics, Washington University School of Medicine, St. Louis, MO, United States of America, 4 University of North Carolina, Chapel Hill, NC, United States of America, 5 Division of Biostatistics, Washington University School of Medicine, St. Louis, Missouri, United States of America, 6 The Charles Bronfman Institute for Personalized Medicine, Icahn School of Medicine at Mount Sinai, New York, NY, United States of America, 7 MRC Human Genetics Unit, MRC Institute of Genetics and Molecular Medicine, University of Edinburgh, Edinburgh, United Kingdom, 8 Jackson Heart Study, University of Mississippi Medical Center, MS, United States of America, 9 Department of Internal Medicine, Section of Gerontology and Geriatrics, Leiden University Medical Center, Leiden, the Netherlands, 10 Section of Hospital Medicine, Veterans Affairs Eastern Colorado Healthcare System, Denver, CO, United States of America, 11 Division of General Internal Medicine, University of Colorado School of Medicine, Aurora, CO, United States of America, 12 Colorado Cardiovascular Outcomes Research Consortium, Aurora, CO, United States of America, 13 Division of Preventive Medicine, Brigham and Women's Hospital, Boston, MA, United States of America, 14 Icelandic Heart Association, Kopavogur, Iceland, 15 Faculty of Medicine, University of Iceland, Reykjavik, Iceland, 16 Laboratory of Epidemiology and Population Science, National Institute on Aging, National Institutes of Health, Baltimore, MD, United States of America, 17 Department of Epidemiology, Erasmus University Medical Center, Rotterdam, The Netherlands, 18 Dean's Office, University of Kentucky College of Public Health, Lexington, Kentucky, United States of America, 19 Diabetes Unit and Center for Genomic Medicine, Massachusetts General Hospital, Massachusetts General Hospital, Boston, MA, United States of America, 20 Programs in Metabolism and Medical & Population Genetics, Broad Institute, Cambridge, MA, United States of America, 21 Department of Medicine, Harvard Medical School, Boston, MA, United States of America, 22 Department of Genetics, School of Medicine, Mashhad University of Medical Sciences, Mashhad, Iran, 23 Laboratory of Epidemiology and Population Sciences, National Institute on Aging, Intramural Research Program, National Institutes of Health, Bethesda, MD, United States of America, 24 Division of Public Health Sciences, Fred Hutchinson Cancer Research Center, Seattle, WA, United States of America, 25 Department of Clinical Epidemiology, Leiden University Medical Center, Leiden, The Netherlands, 26 Department of Public Health and Primary Care, Leiden University Medical Center, Leiden, The Netherlands, 27 The Institute of Medical Sciences, Aberdeen Biomedical Imaging Centre, University of Aberdeen, Aberdeen, United Kingdom, 28 Laboratory of Neurogenetics, National Institute on Aging, National Institutes of Health, Bethesda, MD, United States of America, 29 Data Tecnica International LLC, Glen Echo, MD, United States of America, 30 Department of Internal Medicine, Erasmus University Medical Center, Rotterdam, The Netherlands, 31 Human Genomics Laboratory, Pennington Biomedical Research Center, Louisiana State University System, Baton Rouge, LA, United States of America, 32 Harvard Medical School, Boston, MA, United States of America, 33 Department of Medicine, University of Mississippi Medical Center, Jackson, MS, United States of America, 34 Icelandic Heart Association, Kopavogur, Iceland, 35 University of Iceland, Reykjavik, Iceland, 36 Department of Medicine, School of Medicine, Johns Hopkins University, Baltimore, MD, United States of America, 37 Welch

**Data Availability Statement:** Our study data are now available at the following URL on the AMP T2D Knowledge Portal: http://www.kp4cd.org/dataset_downloads/t2d.

**Funding:** WHI program is funded by the National Heart, Lung, and Blood Institute, National Institutes of Health, U.S. Department of Health and Human Services through contracts HHSN268201100046C, HSN268201100001C, HHSN268201100002C, HHSN268201100003C, HHSN268201100004C, and HHSN271201100004C. The grant funding of WHI are R21 HL123677, R56 DK104806 and R01 MD012765 to NF. The FamHS was funded by R01HL118305 and R01HL117078 NHLBI grants, and 5R01DK07568102 and 5R01DK089256 NIDDK grant." and "The Healthy Aging in Neighborhoods of Diversity across the Life Span (HANDLS) study was supported by the Intramural Research Program of the National Institute on Aging, National Institutes of Health (project # Z01-AG000513 and human subjects protocol number 09-AGN248). Support for GENOA was provided by the National Heart, Lung and Blood Institute (HL119443, HL087660, HL054464, HL054457, and HL054481) of the National Institutes of Health. Ruth loos is supported by the NIH (R01DK110113, U01HG007417, R01DK101855, R01DK107786). The Rotterdam Study GWAS datasets are supported by the Netherlands Organisation of Scientific Research NWO Investments (nr. 175.010.2005.011, 911-03-012), the Research Institute for Diseases in the Elderly (014-93-015; RIDE2), and the Netherlands Genomics Initiative (NGI)/Netherlands Organisation for Scientific Research (NWO) Netherlands Consortium for Healthy Aging (NCHA), project nr. 050-060-810. The ERF study as a part of EUROSPAN (European Special Populations Research Network) was supported by European Commission FP6 STRP grant number 018947 (LSHG-CT-2006- 01947) and also received funding from the European Community's Seventh Framework Programme (FP7/2007-2013)/grant agreement HEALTH-F4-2007-201413 by the European Commission under the programme "Quality of Life and Management of the Living Resources" of 5th Framework Programme (no. QLG2-CT-2002- 01254). The ERF study was further supported by ENGAGE consortium and CMSB. Highthroughput analysis of the ERF data was supported by joint grant from Netherlands Organisation for Scientific Research and the Russian Foundation for Basic Research (NWORFBR 047.017.043).ERF was further supported by the ZonMw grant (project 91111025), and this work was partially supported by the National Heart, Lung and Blood Institute's Framingham Heart Study (Contract No. N01-HC25195) and its contract with Affymetrix, Inc for genotyping services (Contract No. N02-HL-6-4278). This study is also supported by National Institute for Diabetes and Digestive and Kidney

Center for Prevention, Epidemiology and Clinical Research, Johns Hopkins University, Baltimore, MD, United States of America, **38** Department of Epidemiology, Bloomberg School of Public Health, Johns Hopkins University, Baltimore, MD, United States of America, **39** The Mindich Child Health and Development Institute, Ichan School of Medicine at Mount Sinai, New York, NY, United States of America, **40** Departments of Medicine, Brigham and Women's Hospital, Boston, MA, United States of America, **41** Beth Israel Deaconess Medical Center, Boston, MA, United States of America, **42** The Institute for Translational Genomics and Population Sciences, Department of Pediatrics, The Lundquist Institute for Biomedical Innovation at Harbor-UCLA Medical Center, Torrance, CA, United States of America, **43** The New York Academy of Medicine, New York, NY, United States of America, **44** Division of Population Health and Genomics, University of Dundee, Dundee, United Kingdom, **45** Division of Biostatistics, Washington University School of Medicine, St. Louis, MO, United States of America, **46** The National Heart, Lung, and Blood Institute's Framingham Heart Study, Framingham, MA, United States of America, **47** Division of General Internal Medicine Division, Massachusetts General Hospital, Boston, MA, United States of America, **48** Department of Medicine, Harvard Medical School, Boston, MA, United States of America, **49** VA Boston Healthcare System, Boston, MA, United States of America

† Deceased.
* peitaowu@bu.edu (PW); jvassy@partners.org (JLV); ctliu@bu.edu (CTL)

## Abstract

Smoking is a potentially causal behavioral risk factor for type 2 diabetes (T2D), but not all smokers develop T2D. It is unknown whether genetic factors partially explain this variation. We performed genome-environment-wide interaction studies to identify loci exhibiting potential interaction with baseline smoking status (ever vs. never) on incident T2D and fasting glucose (FG). Analyses were performed in participants of European (EA) and African ancestry (AA) separately. Discovery analyses were conducted using genotype data from the 50,000-single-nucleotide polymorphism (SNP) ITMAT-Broad-CARe (IBC) array in 5 cohorts from from the Candidate Gene Association Resource Consortium (n = 23,189). Replication was performed in up to 16 studies from the Cohorts for Heart Aging Research in Genomic Epidemiology Consortium (n = 74,584). In meta-analysis of discovery and replication estimates, 5 SNPs met at least one criterion for potential interaction with smoking on incident T2D at $p<1×10^{-7}$ (adjusted for multiple hypothesis-testing with the IBC array). Two SNPs had significant joint effects in the overall model and significant main effects only in one smoking stratum: rs140637 (*FBN1*) in AA individuals had a significant main effect only among smokers, and rs1444261 (closest gene *C2orf63*) in EA individuals had a significant main effect only among nonsmokers. Three additional SNPs were identified as having potential interaction by exhibiting a significant main effects only in smokers: rs1801232 (*CUBN*) in AA individuals, rs12243326 (*TCF7L2*) in EA individuals, and rs4132670 (*TCF7L2*) in EA individuals. No SNP met significance for potential interaction with smoking on baseline FG. The identification of these loci provides evidence for genetic interactions with smoking exposure that may explain some of the heterogeneity in the association between smoking and T2D.

## Introduction

Cigarette smoking and type 2 diabetes (T2D) are both costly burdens on human health in the United States and worldwide [1–4]. These public health threats are interrelated: smoking is a

Diseases (NIDDK) R01 DK078616 to Drs. Meigs, Dupuis and Florez, NIDDK K24 DK080140 to Dr. Meigs, and a Doris Duke Charitable Foundation Clinical Scientist Development Award to Dr. Florez. The HERITAGE Family Study was supported by National Heart, Lung, and Blood Institute grant HL-45670. The Women's Genome Health Study is supported by the National Heart, Lung, and Blood Instutute (HL043851 and HL080467) and the National Cancer Institute (CA047988 and UM1CA182913). Additional support for endpoint collection was provided by the National Heart, Lung, and Blood Institute under ARRA funding (HL099355). HyperGEN (Hypertension Genetic Epidemiology Network): The hypertension network is funded by cooperative agreements (U10) with NHLBI: HL54471, HL54472, HL54473, HL54495, HL54496, HL54497, HL54509, HL54515, and 2 R01 HL55673- 12. The AGES study has been funded by NIH contracts N01-AG-1-2100 and 271201200022C. Caroline Hayward is supported by an MRC University Unit Programme Grant MC_UU_00007/10 (QTL in Health and Disease)"and "Generation Scotland received core funding from the Chief Scientist Office of the Scottish Government Health Directorate CZD/16/6, the Scottish Funding Council HR03006 and the Wellcome Trust through a Strategic Award (reference 104036/Z/14/Z) for Stratifying Resilience and Depression Longitudinally (STRADL). Genotyping was funded by the UK's Medical Research Council. Jose C. Florez, NIDDK K24 DK110550 The MESA project is conducted and supported by the National Heart, Lung, and Blood Institute (NHLBI) in collaboration with MESA investigators. Support for MESA is provided by contracts 75N92020D00001, HHSN268201500003I, N01-HC-95159, 75N92020D00005, N01-HC-95160, 75N92020D00002, N01-HC-95161, 75N92020D00003, N01-HC-95162, 75N92020D00006, N01-HC-95163, 75N92020D00004, N01-HC-95164, 75N92020D00007, N01-HC-95165, N01-HC-95166, N01-HC-95167, N01-HC-95168, N01-HC-95169, UL1-TR-000040, UL1-TR-001079, UL1-TR-001420, UL1-TR-001881, and DK063491. Additionally, one or more authors are affiliated with the following commercial entities: Interleukin Genetics, GlaxoSmithKline, Daiichi-Sankyo, AstraZeneca, Data Tecnica International LLC, Illumina Inc., University of California Healthcare, Janssen Pharmaceuticals, Goldfinch Bio, and Novo Nordisk. Please see the Competing Interests Statement for additional details. The funders provided support in the form of salaries for authors but did not have any additional role in the study

dose-dependent risk factor for incident T2D, independent of potential confounders including physical activity and body-mass index (BMI) [5]. Moreover, smoking raises fasting glucose (FG) [6, 7] itself a predictor of incident T2D [8–10]. Experimental studies point to plausible biologic mechanisms through which smoking may directly cause T2D, such as the impairment of insulin-mediated glucose transport [11], insulin sensitivity [12–18], and insulin secretion [19–21].

Not every individual who smokes develops T2D, and the relationship between smoking and T2D has considerable heterogeneity. This variation suggests the possibility of genetic modifiers of the effect of smoking on T2D risk. Genetic studies of smoking behavior [22–27] and T2D and FG [28–36] have separately uncovered hundreds of loci associated with these traits, but no genome-wide association study to date has sought genetic loci that modify the relationships among them. We conducted gene-environment-wide interaction studies (GEWIS) to identify potential gene-by-smoking interactions for both T2D risk and FG among 97,773 cohort study participants of European (EA) and African ancestry (AA).

## Materials and methods

### Study design overview

We conducted two-stage GEWIS analyses to identify potential genotype-smoking interactions for two related traits: incident T2D and baseline FG. Smoking status was dichotomized as individuals who were current or former smokers at baseline (ever smokers) and individuals with no current or past smoking history (never smokers). The discovery stage analyses leveraged data from 5 cohort studies from the Candidate Gene Association Resource (CARe) Consortium. Single-nucleotide polymorphisms (SNPs) that had significant association with a trait in meta-analysis of the discovery cohort data were carried forward for replication in up to 16 cohorts from the Cohorts for Heart & Aging Research in Genomic Epidemiology (CHARGE) Consortium Gene-Lifestyle Interactions Working Group and combined discovery plus replication meta-analysis. The Partners Human Research Committee approved this study.

### Cohort descriptions and sample sizes

In the discovery stage, we analyzed data from five cohorts from the CARe Consortium [37]: The Atherosclerosis Risk in Communities Study (ARIC), the Coronary Artery Risk Development in Young Adults Study (CARDIA), the Cardiovascular Health Study (CHS), the Framingham Heart Study (FHS), and the Multi-Ethnic Study of Atherosclerosis (MESA) (S1 Table) [37]. The total sample size of these five discovery stage cohorts was 23,189, including 18,365 European American (EA) and 4,824 African American (AA). Among 23,189 CARe participants, 10,120 were never smokers and 13,069 were ever smokers, as assessed at their baseline study examinations. In the replication stage, 74,584 individuals from up to 16 cohorts in the Cohorts for Heart & Aging Research in Genomic Epidemiology (CHARGE) Consortium Gene-Lifestyle Interactions Working Group were included, comprised of 61,397 EA participants and 13,187 AA participants. A total of 40,819 and 33,765 were never and ever smokers, respectively (S1 Table) [38]. All five discovery cohorts contributed data for both traits of interest: incident T2D and baseline glucose. Eight replication cohorts contributed data for the incident T2D analyses, and 15 replication cohorts contributed data for the fasting glucose analyses (S1 Table). Across the discovery and replication cohorts, there were 4,040 T2D cases and 48,521 controls among EA participants and 717 cases and 7,180 controls among AA participants.

design, data collection and analysis, decision to publish, or preparation of the manuscript. The specific roles of these authors are articulated in the 'author contributions' section.

**Competing interests:** I have read the journal's policy and the authors of this manuscript have the following competing interests. Dr. Meigs currently has a research grant from GlaxoSmithKline and serves on a consultancy board for Interleukin Genetics. Dr. Florez has received consulting honoraria from Daiichi-Sankyo and AstraZeneca. Dr. Mike A. Nalls is supported by a consulting contract between Data Tecnica International LLC and the National Institute on Aging (NIA), National Institutes of Health (NIH), Bethesda, MD, USA. Dr. Nalls also consults for Illumina Inc., the Michael J. Fox Foundation, and the University of California Healthcare. DR. Jose C. Florez, Consulting honoraria from Janssen Pharmaceuticals and Goldfinch Bio, and speaking honorarium from Novo Nordisk The other authors declare no conflicts of interest. This does not alter our adherence to PLOS ONE policies on sharing data and materials. There are no patents, products in development or marketed products associated with this research to declare.

## Description of phenotype and covariates

We considered two traits: incident T2D and baseline FG. Presence of T2D was defined by any one of the following criteria: 1) FG $\geq$ 7 mmol/L; 2) on diabetes treatment or HbA1c $\geq$ 6.5%; 3) 2-hr oral glucose tolerance test $\geq$11.1 mmol/L; 4) random/non-fasting glucose $\geq$ 11.1 mmol/L; 5) physician diagnosis of diabetes; or 6) self-reported diabetes (**S1 Table**). For the analysis of incident T2D, participants meeting the T2D definition at baseline were excluded. For the remaining participants, time-to-T2D was defined as the time from the date of the baseline examination to the date the T2D case definition was met or, for controls, to the last date of follow-up. For the FG analyses, participants with T2D were excluded, and FG was identified from the baseline measurement taken after a fast of 8 hours or more (**S1 Table**).

## Genotyping

Participants in the CARe Consortium were genotyped with the custom ITMAT-Broad-CARe (IBC) genotyping array (IBC v2 chip), which contains around 50,000 SNPs across 2,000 loci selected for their relationship to cardiovascular disease and its risk factors. Details about SNP selection criteria and genotyping quality control (QC) procedures have been described [39]. Details of the genotyping methods used in the individual CHARGE replication cohorts are presented in **S1 Table**.

## Cohort-level statistical analysis

We performed ancestry-stratified analyses for the two traits within each discovery and replication cohort. Smoking-stratified analyses were also conducted separately in each of the four trait-ancestry combinations. In total, we performed four models for each of four trait-ancestry combinations: an interaction model regressing the trait (incident T2D or FG) on the genetic variant, smoking status, and their interaction term (Model 1); a main effect-only model (Model 2); and two smoking-stratified models, regressing incident T2D or FG on the genetic variant predictor in smokers (Model 3) and nonsmokers (Model 4) separately. All models were covariate-adjusted as described below.

We analyzed incident T2D using Cox proportional hazards models and robust sandwich variance estimators. For cohorts with related individuals, each family was treated as a cluster. Models were adjusted for age, BMI, and the genetic principal components associated with incident T2D at $p<0.05$. Models were not adjusted for sex in the discovery cohorts due to insufficient numbers of incident T2D cases in all sex/ancestry categories; models were conducted with or without sex adjustment in the replication analyses, depending on the sample size of stratified samples.

For baseline FG, we used linear regression for cohorts with independent samples. For cohorts with family structures, we used generalized estimating equations (GEE) to obtain estimates for Model 1, assuming an exchangeable working correlation matrix, since the GEE model with an interaction term provides robust standard error estimates. Linear mixed effects models were used to evaluate Models 2–4, with random effects to account for family structures. All FG analyses were adjusted for age, sex, BMI and the genetic principal components associated with FG at $p<0.05$.

## Meta-analysis

For both traits, we obtained summary statistics of association from each cohort and then conducted fixed-effect meta-analysis to combine the results. For each trait (incident T2D and FG), we meta-analyzed the results across the cohorts using inverse variance weighting, in EA and AA separately. We defined a potential interaction effect between a locus and smoking if at least one of the following criteria was met: 1) significant SNP-by-smoking interaction; 2) significant

joint 2-degree-of-freedom test of interaction and main effect, excluding SNPs with significant main effects; or 3) significant SNP effect in only one smoking stratum (never or ever smokers). In the discovery stage, significance was defined as $p < 10^{-3}$; we selected all SNPs significant for at least one of these 3 criteria as candidate SNPs. Candidate SNPs were then carried forward for replication in the cohorts of the CHARGE Consortium. We performed meta-analyses with summary statistics from the discovery and replication stages, defining significance as $p < 1 \times 10^{-7}$ for at least one of the 3 criteria above. We selected this significance threshold to conservatively account for multiple hypothesis-testing, since $p < 2 \times 10^{-6}$ is commonly used for studies with the 50,000-SNP IBC genotyping array [40, 41] and we performed a total of 20 tests (5×2×2), comprised of 5 models (main effect, interaction effect, joint effect, and 2 smoking stratified analyses) for 2 traits in 2 ancestry groups for each variant.

## Power calculations

Power analyses were performed for a significance level of $\alpha = 1 \times 10^{-7}$ to detect a potential interaction effect on both T2D and FG. For T2D, we approximated the power analysis to detect potential interaction with logistic regression. Under the assumption that the effect size for interaction is similar to the effect size of the main SNP effect, the sample sizes of 4,040 EA cases and 717 AA cases enabled 80% power to detect an odds ratio (OR) of 1.39 in EA and 1.76 in AA, using an unmatched population-based case-control design under an additive genetic model and assuming MAF = 0.3 with 10% T2D prevalence and 30% smoking prevalence. For FG, the sample sizes of 58,783 EA and 17,675 AA enabled 80% power to detect SNPs with $R^2_{GE} \geq 0.06\%$ EA and $\geq 0.2\%$ AA for SNP*interaction effect in interaction testing, using an additive genetic model and assuming variants with $R^2_G = 0.1\%$

## Conditional analysis

We performed conditional analyses for the two significant variants identified in *TCF7L2* in the T2D analysis. In each corhort, we ran the joint (Model 1) and main effect only models (Model 2) described above for rs4132670 conditioned on the most significant variant, rs12243326. The cohort-level conditional analyses were meta-analyzed to obtain overall summary statistics.

## Locus characterization

We queried the National Human Genome Research Institute (NHGR)–European Bioinformatics Institute (EBI) GWAS Catalog for any published trait associations with SNPs achieving GEWIS significance in this study [42]. We also examined the overlap between these SNPs and genomic annotation using HaploReg [43], which collects information from multiple functional annotation resources and reports information about queried SNPs such as genomic position, protein-coding impact, available expression quantitative trait locus (eQTL) data, overlap with known transcription factor binding sites or predicted transcription factor binding motifs, and overlap with DNAse hypersensitivity sites or histone marks associated with promoters and enhancers. In addition, we queried each GEWIS-significant SNP in RegulomeDB [44], a database of known and predicted regulatory elements in human intergenic regions, and in the Genotype-Tissue Expression project (GTEx) portal to obtain additional eQTL data [45].

## Results

### Incident T2D

A total of 371 SNPs met the $p < 10^{-3}$ threshold for incident T2D in discovery stage analyses and were carried forward to the replication stage. Of these, 171 were identified among EA

individuals and 200 were identified in AA individuals; no SNP was identified in both sub-groups (**S2 Table**).

In meta-analysis of discovery and replication estimates, five SNPs were significant for potential interaction at $p<1\times10^{-7}$ by at least one criterion, and two of these were significant by two criteria (**Table 1**). Two SNPs had significant joint effects in the overall model and significant main effects in only one smoking stratum in stratified analyses: rs140637 (*FBN1* on chromosome 15, MAF = 0.13) among AA smokers and rs1444261 (closest gene *C2orf63* on chromosome 2, MAF = 0.05) among EA nonsmokers. Among AA participants, rs140637 in *FBN1* was consistently associated with lower T2D risk among smokers only. In the discovery, replication, and combined stage meta-analyses, the per-allele HR for T2D was 0.34 (95% CI = 0.23, 0.51, $p = 8.8$ x $10^{-8}$), 0.39 (95% CI = 0.20, 0.76, $p = 5.3$ x $10^{-3}$), and 0.34 (95% CI = 0.24, 0.49, $p = 2.9$ x $10^{-9}$), respectively. For rs1444261 near *C2orf63*, in the discovery stage, the per-allele hazard ratio (HR) for T2D was 0.64 (95% CI = 0.51, 0.82, $p = 3.7$ x $10^{-4}$) among never smokers, but the direction of effect reversed in the replication stage (HR 1.24, 95% CI = 1.18, 1.29, $p = 3.1$ x $10^{-21}$) and overall meta-analysis (HR 1.21, 95% CI = 1.16, 1.26, $p = 5.1$ x $10^{-18}$).

Three additional SNPs were significant by one criterion only, namely, significant main effect only among smokers in stratified analyses. Among EA smokers, these included rs4132670 (MAF = 0.30) and rs12243326 (MAF = 0.26), both in the well-described T2D-associated gene *TCF7L2*. Among AA smokers, rs1801232, a missense SNP in *CUBN* on chromosome 10 (MAF = 0.12), exhibited a significant main effect (**S1 Fig**). We observed the largest effect size for potential interaction at this *CUBN* missense variant, where the per-allele hazard ratio for T2D was 2.78 (95% CI = 1.92, 4.03, $p = 5.5$ x $10^{-8}$) among smokers and 1.01 (95% CI = 0.58, 1.77, $p = 0.97$) among non-smokers ($p_{joint}$ = 1.3 x $10^{-7}$).

We provide regional plots for rs1224336 in *TCF7L2* in **Fig 1** because the discovery stage, replication stage, and combined meta-analysis showed chip-wide significance for joint effect and main effect in smokers among EA participants. Among smokers and non-smokers, the per-allele HR for T2D in the discovery plus replication meta-analysis was 0.90 (95% CI = 0.86, 0.93, $p = 3.2$ x $10^{-8}$) and 0.96 (95% CI = 0.94, 0.98, $p = 7.5$ x $10^{-5}$), respectively. In analyses conditioned on rs12243326, rs4132670 ($r^2$ = 0.72 and D' = 0.95) was no longer significantly associated with main effect with T2D (all $p>0.4$).

## Fasting glucose

In the discovery stage analysis for baseline FG among 23,189 participants, we observed 343 SNPs meeting the significance threshold of $p<10^{-3}$ in at least one of the three planned strategies for potential interaction: 175 among EA participants and 168 among AA participants. Again, no locus was identified in both ancestral subgroups (**S3 Table**). Meta-analysis identified rs4132670 in *TCF7L2* (MAF = 0.30) as the most significant variant for the joint effect analysis in EA participants only ($p = 4.6$ x$10^{-8}$), but it did not meet the criteria for potential interaction because its main effect association was also significant ($p = 2.8 \times10^{-10}$)

## Locus characterization

Of the five SNPs at four loci achieving statistical significance in the GEWIS analyses (*TCF7L2*, *CUBN*, *FBN1*, and near *C2orf63*), only rs12243326, an intronic variant in *TCF7L2*, has trait associations in the NHGRI-EBI GWAS Catalog, with the glycemic traits of 2-hour glucose challenge, fasting insulin, FG, and BMI interaction on FG. Of the five SNPs at four loci achieving statistical significance in the GEWIS analyses (*TCF7L2*, *CUBN*, *FBN1*, and near *C2orf63*), only the missense *CUBN* SNP is a nonsynonymous variant. All five GEWIS-significant SNPs

**Table 1. Results of discovery (D), replication (R), and combined (D+R) stage meta-analyses of genotype-by-ever smoking for incident type 2 diabetes (T2D).** Bold text indicates a significant potential interaction effect between a SNP and smoking by at least one of the following criteria: (1) significant SNP-by-smoking interaction (p_int); (2) significant joint 2 degree of freedom test of interaction and main effect, excluding SNPs with significant main effects (p_joint); or (3) significant SNP effect in only one smoking stratum (ever or never smokers, p_ever or p_never). No locus met D+R significance at $p < 10^{-7}$ for association with baseline fasting glucose.

| Trait | Race | SNP | CHr | Position | A1 | A2 | Freq1 | Closest gene | Stage | beta_main | p_main | beta_int | p_int | p_joint | beta_ever | p_ever | beta_never | p_never |
|---|---|---|---|---|---|---|---|---|---|---|---|---|---|---|---|---|---|---|
| T2D | EA | rs1444261 | 2 | 55207970 | T | C | 0.95 | C2orf63 | D | -2.00E-01 | 1.8E-02 | 3.30E-01 | 5.3E-02 | 2.8E-03 | 8.00E-02 | 4.8E-01 | -4.40E-01 | **3.7E-04** |
| | | | | | | | | | R | 3.90E-03 | 8.4E-01 | -1.87E-01 | 1.6E-04 | 1.6E-23 | -1.64E-02 | 7.1E-01 | 2.11E-01 | 3.1E-21 |
| | | | | | | | | | D+R | -6.80E-03 | 7.3E-01 | -1.47E-01 | 2.0E-03 | **2.5E-20** | -2.44E-02 | 5.5E-01 | 1.90E-01 | **5.1E-18** |
| T2D | EA | rs4132670 | 10 | 114757761 | A | G | 0.30 | TCF7L2 | D | 2.30E-01 | 4.0E-07 | 6.90E-03 | 9.4E-01 | 2.8E-06 | 2.41E-01 | **1.5E-05** | 2.23E-01 | 3.5E-03 |
| | | | | | | | | | R | 5.30E-02 | 5.4E-09 | 2.27E-02 | 3.1E-01 | 2.7E-09 | 9.89E-02 | 9.9E-07 | 4.15E-02 | 4.5E-05 |
| | | | | | | | | | D+R | 6.00E-02 | 1.7E-11 | 2.18E-02 | 3.2E-01 | 1.3E-12 | 1.16E-01 | **9.6E-10** | 4.48E-02 | 8.9E-06 |
| T2D | EA | rs12243326 | 10 | 114778805 | T | C | 0.74 | TCF7L2 | D | -2.54E-01 | 3.7E-08 | -1.24E-02 | 8.9E-01 | 2.3E-07 | -2.71E-01 | **1.5E-06** | -2.21E-01 | 4.9E-03 |
| | | | | | | | | | R | -4.84E-02 | 2.6E-07 | -1.50E-02 | 5.1E-01 | 3.5E-07 | -8.45E-02 | 4.4E-05 | -3.80E-02 | 3.1E-04 |
| | | | | | | | | | D+R | -5.67E-02 | 7.1E-10 | -1.48E-02 | 5.0E-01 | 1.3E-10 | -1.07E-01 | **3.2E-08** | -4.13E-02 | 7.5E-05 |
| T2D | AA | rs1801232 | 10 | 16910918 | T | G | 0.12 | CUBN | D | 7.77E-01 | 8.2E-06 | 6.95E-01 | 1.1E-01 | **4.7E-07** | 9.67E-01 | **5.0E-07** | 2.72E-01 | 4.9E-01 |
| | | | | | | | | | R | 1.20E-03 | 9.9E-01 | 1.39E+00 | 6.4E-02 | 1.7E-01 | 1.29E+00 | 4.2E-02 | -3.12E-01 | 4.6E-01 |
| | | | | | | | | | D+R | 6.24E-01 | 6.4E-05 | 8.64E-01 | 2.0E-02 | 1.3E-07 | 1.02E+00 | **5.5E-08** | 9.70E-03 | 9.7E-01 |
| T2D | AA | rs140637 | 15 | 46554147 | A | G | 0.87 | FBN1 | D | -6.38E-01 | 1.4E-03 | -1.27E+00 | 6.0E-03 | **2.8E-06** | -1.07E+00 | **8.8E-08** | 1.25E-01 | 7.6E-01 |
| | | | | | | | | | R | -5.27E-01 | 1.6E-02 | -7.25E-01 | 1.3E-01 | 6.9E-03 | -9.41E-01 | 5.3E-03 | 5.40E-03 | 9.9E-01 |
| | | | | | | | | | D+R | -5.88E-01 | 6.7E-05 | -1.01E+00 | 2.2E-03 | **2.2E-08** | -1.07E+00 | **2.9E-09** | 5.49E-02 | 8.3E-01 |

Abbreviations: A: allele, AA: African-American, Chr: chromosome, EA: European-American. Freq1: allele frequency of the coded effect allele (A1).

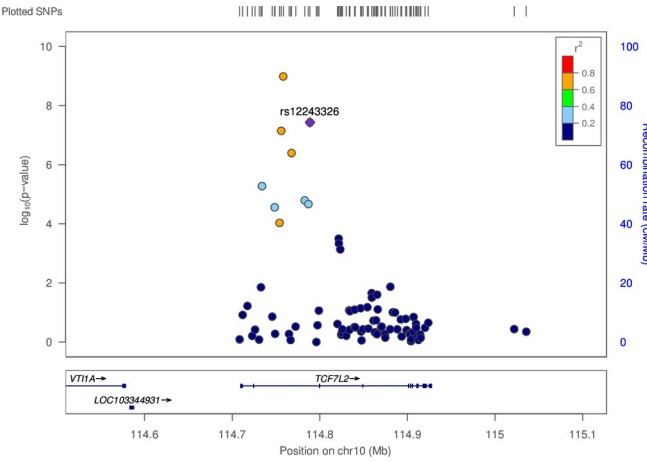

**a. Main effect**

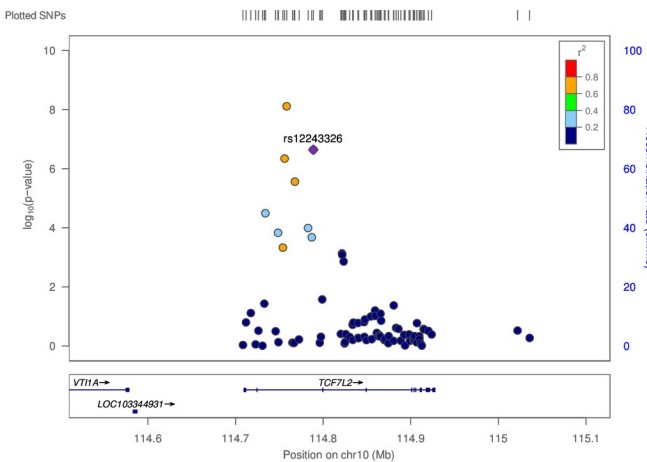

**b. Joint effect**

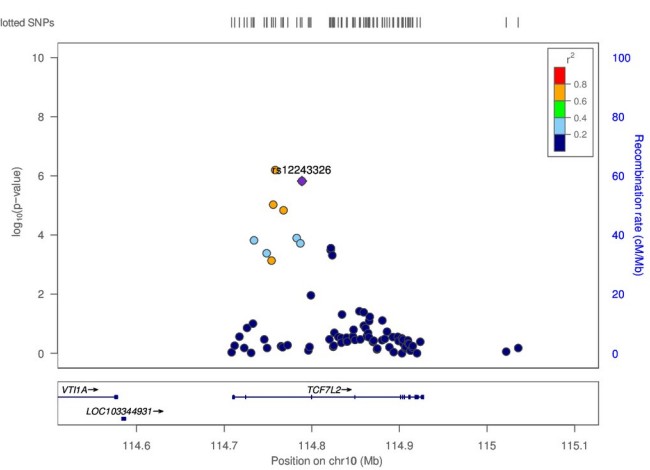

**c. Main effect in smokers only**

**Fig 1. Regional plots for the association of rs1224336 in *TCF7L2* with T2D.**

overlap with at least one promoter or enhancer regulatory mark in at least one tissue with relevance to diabetes, including brain, muscle, gastrointestinal tract, pancreas, adipose, and liver (**S4 Table**). SNPs at three of the four loci (*C2orf63*, *TCF7L2*, and *CUBN*) had eQTL associations, and SNPs at all four loci overlap with either a DNA-binding site or alter a predicted DNA-binding motif (**S4 Table**).

## Discussion

Using data from 61,164 participants from 19 cohort studies, we performed two GEWIS to identify potential SNP-by-smoking interactions in the risk of T2D and baseline FG. We identified potential interactions between smoking status and five SNPs at or near four genes (*TCF7L2*, *CUBN*, *C2orf63 (closest gene)*, and *FBN1*) on the risk of incident T2D in EA or AA participants. We identified no significant SNP-smoking interactions for FG.

The relationship between smoking and T2D is complex and likely results from both confounding and true causal relationships [46]. Smokers are less likely to be physically active [47]

and more likely to have unhealthier dietary intake [48, 49]. Still, a meta-analysis of 25 prospective studies by Willi found that smokers had a risk ratio for incident T2D of 1.44 (95% CI 1.31, 1.58) over 5 to 30 years of follow-up after adjustment, when possible, for BMI, physical activity, and other potential confounders. Individuals with the greatest smoking exposure had the greatest T2D risk [5]. Moreover, experimental data suggest plausible causal pathways between smoking and T2D. First, smoking generates reactive oxygen species (ROS) [50], which decrease *in vitro* insulin-mediated glucose transport [11]. Second, smoking stimulates the sympathetic system and cortisol release, increasing central obesity and insulin resistance [12–14]. Nicotine may mediate these pathways, as it increases insulin resistance [15–18], possibly through increased ROS production and TNF-α expression [18]. Nicotine also decreases insulin secretion from pancreatic β-cells [19], and fetal and neonatal exposure to nicotine results in β-cell dysfunction and apoptosis [20,21]. GEWIS might help elucidate additional biological pathways to explain the relationship between smoking and T2D. A linkage disequilibrium regression score study of 276 genetic correlations among 24 traits found no genetic correlation between smoking status and either T2D or FG [51], but one small study has reported that smoking status accounted for 22% of the gene-environment variance in β-cell function, as measured by the homeostatic model assessment (HOMA-β) [52].

We observed the largest potential interaction effect size at the missense SNP rs1801232 in the *CUBN* gene in individuals of African ancestry, where the per-allele hazard ratio for T2D was 2.78 (95% CI = 1.92, 4.03, $p = 5.5 \times 10^{-8}$) among smokers and 1.01 (95% CI = 0.58, 1.77, $p = 0.97$) among non-smokers ($p_{joint} = 1.3 \times 10^{-7}$). Cubilin is a component of the vitamin B12-intrinsic factor complex receptor in the ileal mucosa [53], and it is expressed in the apical brush border of the renal proximal tubule, where it participates in receptor-mediated endocytosis of low-molecular-weight proteins [54]. Defects in the *CUBN* gene have been associated with both vitamin B12 deficiency and proteinuria, and the absence of cubilin results in the autosomal recessive condition Imerslund-Gräsbeck syndrome, characterized by B12 malabsorption and variable levels of proteinuria from impaired renal protein reabsorption [55]. Mice heterozygous for *CUBN* deletion have increased albuminuria and decreased levels of blood albumin and high-density lipoprotein (HDL) cholesterol [56]. The CKDGen consortium meta-analysis identified a missense SNP in *CUBN* (rs18801239) associated with urinary albumin/creatinine ratio and clinical microalbuminuria in the general population, an association replicated in an AA cohort with type 1 diabetes [57] and later in the Framingham Offspring Study [58]. This SNP appears independent from the *CUBN* SNP identified in the present analysis: in conditional analyses on rs18801239 in the discovery cohort, we found that rs18801232 remained significantly associated with incident T2D among AA smokers only. These *CUBN* observations point to plausible mechanisms, namely depressed levels of vitamin B12 and HDL cholesterol, through which smoking might interact with cubilin to cause T2D. Cigarette smoking impairs cubilin-mediated renal protein reabsorption through cadmium and other contaminants, which form complexes with proteins that have high affinity for cubilin and accumulate in the proximal tubule [59]. A mendelian randomization study found an association between a genetic instrument for low vitamin B12 levels (including one *CUBN* variant) and higher fasting glucose levels and lower pancreatic beta-cell secretory function, as measured by HOMA-β, but not with higher odds of T2D [60]. Mendelian randomization studies have been inconsistent in whether genetic instruments for low HDL are associated with increased T2D risk [61–64]. Whether *CUBN* defects and smoking interact to cause T2D through these or other mechanisms merits further investigation.

We observed more modest potential interaction effects at four other SNPs. Among AA participants, one SNP in *FBN1* was associated with T2D only in smokers. The glycoprotein fibrillin-1 is a component of microfibrils in the extracellular matrix, which contribute to the

elasticity of skin, blood vessels, and other tissues. Variants in *FBN1* are associated with Marfan syndrome, an autosomal dominant connective tissue disorder characterized by ocular, skeletal, and cardiovascular abnormalities, including aortic dilatation and cardiac valve regurgitation [65]. Among EA participants, one locus near *C2orf63*, which encodes a neurite outgrowth inhibitor, was associated with T2D only in never smokers. This observation may suggest either a protective role of smoking in the association of *C2orf63* and T2D or an *C2orf63*-T2D association otherwise obscured by the association between smoking and T2D. The two remaining loci we identified were in *TCF7L2*, a gene whose well-established association with T2D was first identified in 2006 and which remains the locus with the largest effect on T2D risk [66–68]. Variants in *TCF7L2* are associated with decreased pancreatic beta-cell function [69,70] and incretin sensitivity [71], and their association with increased proinsulin levels suggest defects in insulin processing and secretion [72]. Experimental models support the role of *TCF7L2* variants in developmental beta cell proliferation, proinsulin processing, and insulin vesicle docking [73].

Examination of the functional genomic annotation of the GEWIS-significant SNPs generates novel biological hypotheses. For example, allele-specific differential gene expression impacting glucose homeostasis in smokers versus non-smokers could explain the observed potential gene-smoking interaction. A mechanism of interaction involving gene expression would be consistent with all five statistically-significant SNPs being associated with regulatory histone marks. Even the missense variant in the *CUBN* gene overlaps with regulatory annotation in numerous tissues, including active enhancer histone marks in muscle, adipose, pancreas, and liver, and tags multiple DNA-binding protein sites. Similarly, the intergenic SNP at the *C2orf63* locus overlaps with both active enhancer and promoter histone marks from brain/neural tissues. The intronic variant in the *FBN1* gene overlaps with promoter and/or active enhancer marks in brain, muscle, adipose, gastrointestinal tract, or pancreatic tissues. Finally, each of the two intronic SNPs at the *TCF7L2* locus has a slightly different pattern of regulatory annotation. In addition, the pattern of regulatory marks overlapping the two *TCF7L2* SNPs identified in this study differs from the regulatory annotation related to the lead *TCF7L2* SNP associated in T2D case-control GWAS, suggesting multiple, potentially distinct regulatory mechanisms underlying T2D in smokers and non-smokers. Further work is required to illuminate how smoking might modify biologic pathways, including gene regulation, and may suggest novel targets for diabetes therapy.

Prior studies of gene-smoking interaction for T2D risk have used a candidate gene approach, focusing on loci associated either with smoking behavior, such as *CYP2A6* [74] or the *nicotinic acetylcholine receptor gene* (*CHRNA4*) [75], or with T2D and other metabolic traits [76], including *HNF1A* [77] and *APOC3* [78]. Our analyses did not replicate the findings of these small candidate-gene studies at our predefined genome-wide significance thresholds, highlighting unique contributions using unbiased GEWIS approaches. Limitations of our study include the dichotomous categorization of the smoking exposure (ever vs. never), which likely masks some of the effect of smoking dose and duration on our outcomes of interest. Nonetheless, similar approaches have successfully identified gene-smoking interactions for traits such as blood pressure [79], pulmonary function [80], and BMI [81]. Second, a locus identified by the inclusion of a significant joint test as one criterion for potential locus-smoking interaction may actually have a significant main effect, not a significant interaction with smoking, if the inclusion of smoking in the model explained residual variability in the outcome and increased power to detect main effects. To limit the impact of this misclassification, we excluded SNPs with significant main effects from eligibility for this criterion. Third, although we used data from about 75,000 individuals across the CHARGE Consortium Gene-Lifestyle Interactions Working Group to replicate our discovery analyses, data from larger cohorts such

as the UK Biobank and Million Veteran Program now exist and might provide future opportunity for additional replication. Fourth, our discovery analyses only leveraged genotype data from the IBC array available from the CARe Consortium; the use of increasingly available sequencing data from large cohort studies might enable the detection of rare variants that mediate the relationship between smoking and glycemic traits. Fifth, the lack of adequate numbers of T2D cases in all sex/ancestry groups impeded adjustment for sex in some models. It is unknown whether this lack of sex adjustment biased the results and, if so, the direction and magnitude of effect. Larger studies in individuals of non-European ancestry are needed to address this limitation.

## Conclusions

We have demonstrated the feasibility and utility of GEWIS to identify potential gene-smoking interactions in T2D risk. Future mechanistic study of the loci identified may help untangle the complex relationship between the dual public health threats of T2D and smoking.

## Supporting information

**S1 Table. Characteristics of discovery and replication cohorts.**
(XLSX)

**S2 Table. SNPs meeting p<10–3 significance threshold for potential locus-smoking interaction for incident T2D in discovery stage analyses among EA individuals and AA individuals.**
(XLSX)

**S3 Table. SNPs meeting p<10–3 significance threshold for potential locus-smoking interaction for fasting glucose in discovery stage analyses among EA individuals and AA individuals.**
(XLSX)

**S4 Table. Locus characterization of potential locus-smoking interactions for type 2 diabetes risk with publicly available databases.**
(XLSX)

**S1 Fig. Regional plot for rs1801232 with incident type 2 diabetes among smokers of African ancestry, indicating absence of linkage disequilibrium with other SNPs in YRI reference panel from the 1000 Genomes Project.**
(PDF)

## Acknowledgments

A portion of this research utilized the Linux Cluster for Genetic Analysis (LinGA-II) funded by the Robert Dawson Evans Endowment of the Department of Medicine at Boston University School of Medicine and Boston Medical Center.

Genotyping of GENOA was performed at the Mayo Clinic (Stephen T. Turner, MD, Mariza de Andrade PhD, Julie Cunningham, PhD). We thank Eric Boerwinkle, PhD and Megan L. Grove from the Human Genetics Center and Institute of Molecular Medicine and Division of Epidemiology, University of Texas Health Science Center, Houston, Texas, USA for their help with genotyping. We would also like to thank the families that participated in the GENOA study.

The Mount Sinai IPM Biobank Program is supported by The Andrea and Charles Bronfman Philanthropies.

The Rotterdam Study is supported by Erasmus Medical Center and Erasmus University, Rotterdam, Netherlands Organization for the Health Research and Development (ZonMw), the Research Institute for Diseases in the Elderly (RIDE), the Ministry of Education, Culture and Science, the Ministry for Health, Welfare and Sports, the European Commission (DG XII), and the Municipality of Rotterdam.The authors are grateful to the study participants, the staff from the Rotterdam Study and the participating general practitioners and pharmacists. The generation and management of GWAS genotype data for the Rotterdam Study (RS I, RS II, RS III) was executed by the Human Genotyping Facility of the Genetic 20 Laboratory of the Department of Internal Medicine, Erasmus MC, Rotterdam, The Netherlands. The GWAS datasets are additionally supported by the Genetic Laboratory of the Department of Internal Medicine, Erasmus MC. We thank Pascal Arp, Mila Jhamai, Marijn Verkerk, Lizbeth Herrera and Marjolein Peters, MSc, and Carolina Medina-Gomez, MSc, for their help in creating the GWAS database, and Karol Estrada, PhD, Yurii Aulchenko, PhD, and Carolina Medina-Gomez, MSc, for the creation and analysis of imputed data.

The ERF study as a part of EUROSPAN (European Special Populations Research Network) was additionally supported by ENGAGE consortium and CMSB. We are grateful to all study participants and their relatives, general practitioners and neurologists for their contributions and to P. Veraart for her help in genealogy, J. Vergeer for the supervision of the laboratory work, P. Snijders for his help in data collection and E.M. van Leeuwen for genetic imputation.

This research was conducted in part using data and resources from the Framingham Heart Study of the National Heart Lung and Blood Institute of the National Institutes of Health and Boston University School of Medicine. The analyses reflect intellectual input and resource development from the Framingham Heart Study investigators participating in the SNP Health Association Resource (SHARe) project.

The HyperGEN (Hypertension Genetic Epidemiology Network) study involves University of Utah: (Network Coordinating Center, Field Center, and Molecular Genetics Lab); Univ. of Alabama at Birmingham: (Field Center and Echo Coordinating and Analysis Center); Medical College of Wisconsin: (Echo Genotyping Lab); Boston University: (Field Center); University of Minnesota: (Field Center and Biochemistry Lab); University of North Carolina: (Field Center); Washington University: (Data Coordinating Center); Weill Cornell Medical College: (Echo Reading Center); National Heart, Lung, & Blood Institute. For a complete list of Hyper-GEN Investigators:http://www.biostat.wustl.edu/hypergen/Acknowledge.html.

The AGES study is additionally supported by the NIA Intramural Research Program, Hjartavernd (the Icelandic Heart 21 Association), and the Althingi (the Icelandic Parliament).

## Author Contributions

**Data curation:** Denis Rybin, Lawrence F. Bielak, Mary F. Feitosa, Nora Franceschini, Yize Li, Yingchang Lu, Jonathan Marten, Solomon K. Musani, Raymond Noordam, Sridharan Raghavan, Lynda M. Rose, Karen Schwander, Albert V. Smith, Salman M. Tajuddin, Dina Vojinovic, Najaf Amin, Donna K. Arnett, Erwin P. Bottinger, Ayse Demirkan, Jose C. Florez, Mohsen Ghanbari, Tamara B. Harris, Lenore J. Launer, Jingmin Liu, Jun Liu, Dennis O. Mook-Kanamori, Alison D. Murray, Mike A. Nalls, Patricia A. Peyser, André G. Uitterlinden, Trudy Voortman, Claude Bouchard, Daniel Chasman, Adolfo Correa, Renée de Mutsert, Michele K. Evans, Vilmundur Gudnason, Caroline Hayward, Linda Kao, Sharon L. R. Kardia, Charles Kooperberg, Ruth J. F. Loos, Michael M. Province, Tuomo Rankinen, Susan Redline, Paul M. Ridker, Jerome I. Rotter, David Siscovick, Blair H. Smith, Cornelia

van Duijn, Alan B. Zonderman, D. C. Rao, James G. Wilson, Josée Dupuis, James B. Meigs, Ching-Ti Liu, Jason L. Vassy.

**Funding acquisition:** Solomon K. Musani.

**Writing – review & editing:** Peitao Wu, Claude Bouchard.

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
