## [Decision Letter · Decision Letter 0]

27 Nov 2019

PONE-D-19-27852

Smoking-by-Genotype Interaction in Type 2 Diabetes Risk and Fasting Glucose

PLOS ONE

Dear Dr Vassy,

Thank you for submitting your manuscript to PLOS ONE. After careful consideration, we feel that it has merit but does not fully meet PLOS ONE’s publication criteria as it currently stands. Therefore, we invite you to submit a revised version of the manuscript that addresses the points raised during the review process.

We would appreciate receiving your revised manuscript by Jan 11 2020 11:59PM. To enhance the reproducibility of your results, we recommend that if applicable you deposit your laboratory protocols in protocols.io, where a protocol can be assigned its own identifier (DOI) such that it can be cited independently in the future. For instructions see: http://journals.plos.org/plosone/s/submission-guidelines#loc-laboratory-protocols

We look forward to receiving your revised manuscript.

Kind regards,

David Meyre

Academic Editor

PLOS ONE

Journal Requirements:

1. Thank you for including the following funding information within the acknowledgements section of the manuscript; "The WHI program is funded by the National Heart, Lung, and Blood Institute, National Institutes of Health, U.S. Department of Health and Human Services through contracts HHSN268201100046C, HSN268201100001C, HHSN268201100002C, HHSN268201100003C, HHSN268201100004C, and HHSN271201100004C. The grant funding of WHI are R21 HL123677, R56 DK104806 and R01 MD012765 to NF. The FamHS was funded by R01HL118305 and R01HL117078 NHLBI grants, and 5R01DK07568102 and 5R01DK089256 NIDDK grant. " and "The Healthy Aging in Neighborhoods of Diversity across the Life Span (HANDLS) study was supported by the Intramural Research Program of the National Institute on Aging, National Institutes of Health (project # Z01-AG000513 and human subjects protocol number 09-AGN248). Genotyping of GENOA was performed at the Mayo Clinic (Stephen T. Turner, MD, Mariza de Andrade PhD, Julie Cunningham, PhD). We thank Eric Boerwinkle, PhD and Megan L. Grove from the Human Genetics Center and Institute of Molecular Medicine and Division of Epidemiology, University of Texas Health Science Center, Houston, Texas, USA for their help with genotyping. We would also like to thank the families that participated in the GENOA study. Support for GENOA was provided by the National Heart, Lung and Blood Institute (HL119443, HL087660, HL054464, HL054457, and HL054481) of the National Institutes of Health. The Mount Sinai IPM Biobank Program is supported by The Andrea and Charles Bronfman Philanthropies. Ruth loos is supported by the NIH (R01DK110113, U01HG007417, R01DK101855, R01DK107786). The Rotterdam Study is funded by Erasmus Medical Center and Erasmus University, Rotterdam, Netherlands Organization for the Health Research and Development (ZonMw), the Research Institute for Diseases in the Elderly (RIDE), the Ministry of Education, Culture and Science, the Ministry for Health, Welfare and Sports, the European Commission (DG XII), and the Municipality of Rotterdam. The authors are grateful to the study participants, the staff from the Rotterdam Study and the participating general practitioners and pharmacists. The generation and management of GWAS genotype data for the Rotterdam Study (RS I, RS II, RS III) was executed by the Human Genotyping Facility of the Genetic 20 Laboratory of the Department of Internal Medicine, Erasmus MC, Rotterdam, The Netherlands. The GWAS datasets are supported by the Netherlands Organisation of Scientific Research NWO Investments (nr. 175.010.2005.011, 911-03-012), the Genetic Laboratory of the Department of Internal Medicine, Erasmus MC, the Research Institute for Diseases in the Elderly (014-93-015; RIDE2), the Netherlands Genomics Initiative (NGI)/Netherlands Organisation for Scientific Research (NWO) Netherlands Consortium for Healthy Aging (NCHA), project nr. 050-060-810. We thank Pascal Arp, Mila Jhamai, Marijn Verkerk, Lizbeth Herrera and Marjolein Peters, MSc, and Carolina Medina-Gomez, MSc, for their help in creating the GWAS database, and Karol Estrada, PhD, Yurii Aulchenko, PhD, and Carolina Medina-Gomez, MSc, for the creation and analysis of imputed data. The ERF study as a part of EUROSPAN (European Special Populations Research Network) was supported by European Commission FP6 STRP grant number 018947 (LSHG-CT-2006-01947) and also received funding from the European Community's Seventh Framework Programme (FP7/2007-2013)/grant agreement HEALTH-F4-2007-201413 by the European Commission under the programme "Quality of Life and Management of the Living Resources" of 5th Framework Programme (no. QLG2-CT-2002- 01254). The ERF study was further supported by ENGAGE consortium and CMSB. Highthroughput analysis of the ERF data was supported by joint grant from Netherlands Organisation for Scientific Research and the Russian Foundation for Basic Research (NWORFBR 047.017.043). ERF was further supported by the ZonMw grant (project 91111025). We are grateful to all study participants and their relatives, general practitioners and neurologists for their contributions and to P. Veraart for her help in genealogy, J. Vergeer for the supervision of the laboratory work, P. Snijders for his help in data collection and E.M. van Leeuwen for genetic imputation. This research was conducted in part using data and resources from the Framingham Heart Study of the National Heart Lung and Blood Institute of the National Institutes of Health and Boston University School of Medicine. The analyses reflect intellectual input and resource development from the Framingham Heart Study investigators participating in the SNP Health Association Resource (SHARe) project. This work was partially supported by the National Heart, Lung and Blood Institute’s Framingham Heart Study (Contract No. N01-HC25195) and its contract with Affymetrix, Inc for genotyping services (Contract No. N02-HL-6- 4278). A portion of this research utilized the Linux Cluster for Genetic Analysis (LinGA-II) funded by the Robert Dawson Evans Endowment of the Department of Medicine at Boston University School of Medicine and Boston Medical Center. Also supported by National Institute for Diabetes and Digestive and Kidney Diseases (NIDDK) R01 DK078616 to Drs. Meigs, Dupuis and Florez, NIDDK K24 DK080140 to Dr. Meigs, and a Doris Duke Charitable Foundation Clinical Scientist Development Award to Dr. Florez. The HERITAGE Family Study was supported by National Heart, Lung, and Blood Institute grant HL-45670. The Women's Genome Health Study is supported by the National Heart, Lung, and Blood Instutute (HL043851 and HL080467) and the National Cancer Institute (CA047988 and UM1CA182913). Additional support for endpoint collection was provided by the National Heart, Lung, and Blood Institute under ARRA funding (HL099355). HyperGEN (Hypertension Genetic Epidemiology Network): The hypertension network is funded by cooperative agreements (U10) with NHLBI: HL54471, HL54472, HL54473, HL54495, HL54496, HL54497, HL54509, HL54515, and 2 R01 HL55673- 12. The study involves: University of Utah: (Network Coordinating Center, Field Center, and Molecular Genetics Lab); Univ. of Alabama at Birmingham: (Field Center and Echo Coordinating and Analysis Center); Medical College of Wisconsin: (Echo Genotyping Lab); Boston University: (Field Center); University of Minnesota: (Field Center and Biochemistry Lab); University of North Carolina: (Field Center); Washington University: (Data Coordinating Center); Weil Cornell Medical College: (Echo Reading Center); National Heart, Lung, & Blood Institute. For a complete list of HyperGEN Investigators: http://www.biostat.wustl.edu/hypergen/Acknowledge.html. The AGES study has been funded by NIH contracts N01-AG-1-2100 and 271201200022C, the NIA Intramural Research Program, Hjartavernd (the Icelandic Heart 21 Association), and the Althingi (the Icelandic Parliament). "

"Please see Acknowledgements statement in manuscript."

2. Thank you for including your competing interests statement; "Please see Competing Interests section in manuscript."

4. Your ethics statement must appear in the Methods section of your manuscript. If your ethics statement is written in any section besides the Methods, please move it to the Methods section and delete it from any other section. Please also ensure that your ethics statement is included in your manuscript, as the ethics section of your online submission will not be published alongside your manuscript.

Reviewers' comments:

Reviewer's Responses to Questions

**Comments to the Author**

1. Is the manuscript technically sound, and do the data support the conclusions?

Reviewer #1: Partly

Reviewer #2: Partly

Reviewer #3: Partly

2. Has the statistical analysis been performed appropriately and rigorously? 

Reviewer #1: Yes

Reviewer #2: Yes

Reviewer #3: Yes

3. Have the authors made all data underlying the findings in their manuscript fully available?

Reviewer #1: No

Reviewer #2: Yes

Reviewer #3: Yes

4. Is the manuscript presented in an intelligible fashion and written in standard English?

Reviewer #1: Yes

Reviewer #2: Yes

Reviewer #3: Yes

5. Review Comments to the Author

Reviewer #1: This is a paper investigating smoking by genotype interaction in incident T2DM and FG using data from two well-known consortia. The paper is well written and addresses a relevant topic.

There is something about this manuscript that puzzles me. In line 146-158 the authors briefly describe the cohorts from the two consortia contributing to this study, referring to Supplemental Table 1. According to the text, data from 5 cohorts were used for the discovery stage, and data from 14 cohorts in the replication stage.

In Supplemental Table 1, however, I noticed that for the replication stage, only 4 out of 14 cohorts contribute to the GEWIS for incident type 2 diabetes (row 12 in Supplemental Table 1: 10 do not have data on incident T2DM). In addition, only 12 cohorts contribute to the GEWIS for fasting glucose (2 do not have glucose measurements). This seemed odd to me, because the significant interactions observed in this study are for the analyses concerning incident T2DM, not for fasting glucose.....

The authors do not acknowledge this fact (4 instead of 14 cohorts for incident T2D in replications stage) anywhere in the manuscript. There may have been a mistake, an old version of Supplemental Table 1 may have been uploaded, or the results of the replications stage are really based on only 4 resp. 12 cohorts. If the latter is the case, then the authors should describe this early in the manuscript, because it puts the results in a completely different perspective. Now the authors only refer to their small number of incident T2D cases in the context of not being able to adjust for sex (line 197 and 422).

Have the authors performed a power analyses before conducting the study? If so, please include.

In the cohort description it would really help to also list the number of incident cases of type 2 diabetes for the discovery and replication stage cohorts. It would also be informative to include the number of cohorts with family structures in discovery and replications stages.

Why did the authors include the results from both the discovery and replication estimates in the meta-analysis, why not restrict this to replication results only?

In the discussion a paragraph needs to be included about how the results and their meaning/interpretation are affected by the fact that sex could not be taken into account in most analyses.

Minor comments

Regulome DB is not included in the methods section, but is included in Suppl. T4

In line 269 please prove the 95% CI

In line 302 is reference is made to Table 1, but glucose results are not included in this Table.

For example, the refs 24 and 25 in line 337 are incorrect (refs 24 and 25 in the list (p. 23) do not cover the topic of B-cell dysfunction and apoptosis)

Reviewer #2: Wu et al. have performed an array-wide association study for gene-smoking interactions on fasting glucose and incident T2D. They do not identify significant interaction effects, but report some novel significant associations when testing joint associations with SNP main effect and interaction, and associations that reach the significance in one stratum (smokers or ever smokers) but not the other. The paper includes some novel findings, but I have concerns about how the findings have been reported.

1. Defining interaction as a SNP main effect that is significant in only one smoking stratum (never or ever smokers) is misleading, as this does not give any information of how the SNP effect differs between the strata, i.e. the interaction effect. Please do not define these findings as interaction effects, but use appropriate wording.

2. It is also misleading to define the joint association of SNP main effect and interaction effect as an interaction. The identification of a locus in the joint test but not in a SNP main effect test could be due to an interaction effect, but could also be simply due to the adjustment of the model for smoking which increases power by explaining some of the residual variability in the outcome trait.

3. Lines 214-215: “…excluding SNPs with significant main effects…” What P value was considered a significant SNP main effect? Considering that the TCF7L2 locus is the locus with the strongest known main effect on T2D risk, it is surprising that TCF7L2 was not excluded at this stage.

4. Please do not only report results for single SNPs but indicate clearly when the SNPs represent independent loci, based on distance and/or LD threshold values, or conditional analyses.

5. Lines 267-271: It is not possible that a SNP has a significant T2D-risk decreasing effect HR=0.64 in the discovery stage and a significant risk-increasing effect in the replication stage with HR=1.19, and the combined meta-analysis P value is still significant with HR=1.17. Please double-check this result. Due to the opposite direction of effect between the discovery and replication stages, the combined P value should be close to HR=1.

6. Line 269: “CI=XXX, XXX”, please add the missing numbers here.

7. Line 355-356: Important to clarify whether the CKDGen consortium meta-analysis identified the same CUBN missense SNP as identified in the present study.

8. Line 420: “such as” words are out of place here.

9. Lines 426-428: The conclusions are not justified by the data. This paragraph needs to be revised to be consistent with the findings. It should be clearly stated that no significant interaction effects were found. Rather, the authors found some loci that showed significant joint test associations for SNP main effect and interaction, and associations that were significant in one smoking stratum but not the other. Future studies with larger sample sizes are required to provide evidence of interaction for these loci.

10. TABLE 1: It is critical to indicate which allele is the effect allele, and to add effect allele frequency in the table.

11. FIGURE 1d: The association pattern looks strange. What was the sample size and imputation quality for this variant? What reference panel was used for the LD?

Reviewer #3: Wu et al. describe results from Gene x Smoking interactions analyses for type 2 diabetes (T2D) and fasting glucose (FG).

The study is based on a 2-stage design, including a discovery and replication stage. The discovery stage consists of 5 cohorts from the CARe consortium (N~24,000). The discovery samples were genotyped on the Custom ITMAT-Broad-CARe (IBC) genotyping array that contains ~50K SNPs selected for their impact on cardiovascular diseases. The replication stage consists of data from the CHARGE consortium (N~75,000). Both stages involve individuals of European and African ancestry (EA and AA, respectively). All analyses were stratified by ancestry status. Smoking was defined by a binary variable SMK that compares ever and never smokers.

For each ancestry group, each trait and each of the two stages, the authors applied four regression models: (i) An interaction model that includes SNP, SMK, SNPxSMK and other covariates, (ii) a main effect model (excluding the interaction term), (iii) a main effect model that was restricted to ever smokers and (iv) a main effect model that was restricted to never smokers. For each stage, the authors conducted fixed-effect meta-analysis to combine cohort-specific results. In addition, the results of the individual stages were meta-analysed.

They selected variants from discovery that showed (a) significant SNP x Smk interaction (Pint < 1e-3), (b) significant joint main+interaction effect (Pjoint <1e-3, excluding those with significant main effects Pmain <1e-3), or (c) significant subgroup effects (P<1e-3, excluding those that show an effect in the other subgroup as well). They selected 371 variants from discovery, which were then followed-up in a combined discovery+replication stage. They identified 6 SNPs for T2D with P<1e-7 for at least one of the three approaches (a)-(c). They did not identify any variant for FG. The identified variants are likely candidates for SNP x Smk interactions for T2D.

The study is informative and adds to previous work. It provides additional insight into the biological and genetic underpinning of T2D. The results will be of interest for those studying diabetes and related traits. I only have some minor comments:

1. Table 1 shows results for the 6 variants. It seems that only three of the 6 variants have nominal significant interaction P values. I think this deserves some discussion as to how much one can trust those interactions. Also, I would find it helpful to add P values for the test for difference between beta_never and beta_ever that may help the interpretation of these findings. The abstract says that the findings “provide evidence” for interactions. Personally, I would tone down on that a little bit. No doubt, the identified loci are likely candidates for interactions but not necessarily proven (as mentioned above, P int is not even nominal significant for three of the 6 variants).

2. Have the authors thought of testing the interaction for known main effect T2D and FG variants?

3. Page 3, line 100: “cohorts”

4. Page 11, line 269: CI’s are missing

6. PLOS authors have the option to publish the peer review history of their article (what does this mean?). If published, this will include your full peer review and any attached files.

Reviewer #1: No

Reviewer #2: No

Reviewer #3: No

---

## [Author Response · Author response to Decision Letter 0]

3 Mar 2020

Please see full response to reviewers in the attached Cover Letter.

---

## [Editor Report · Decision Letter 1]

10 Mar 2020

Smoking-by-Genotype Interaction in Type 2 Diabetes Risk and Fasting Glucose

PONE-D-19-27852R1

Dear Vassy,

We are pleased to inform you that your manuscript has been judged scientifically suitable for publication and will be formally accepted for publication once it complies with all outstanding technical requirements.

With kind regards,

David Meyre

Academic Editor

PLOS ONE
---

## [Editor Report · Acceptance letter]

20 Apr 2020

PONE-D-19-27852R1 

Smoking-by-Genotype Interaction in Type 2 Diabetes Risk and Fasting Glucose 

Dear Dr. Vassy:

I am pleased to inform you that your manuscript has been deemed suitable for publication in PLOS ONE. Congratulations! Your manuscript is now with our production department. 

With kind regards,

on behalf of

Dr David Meyre 

Academic Editor

PLOS ONE